# Hypergammaglobulinemia in treated and untreated people with HIV

Isabel Vigmo[1,2]*, Staffan Nilsson[3], Aylin Yilmaz[1,4], Magnus Gisslén[1,4], Josefina Robertson[1,4☯], Johanna Karlsson[1,2☯]

1 Department of Infectious Diseases, Institute of Biomedicine, Sahlgrenska Academy, University of Gothenburg, Gothenburg, Sweden, 2 Department of Infectious Diseases, NU Hospital Group, Trollhättan, Region Västra Götaland, Sweden, 3 Department of Laboratory Medicine, Institute of Biomedicine, Sahlgrenska Academy, University of Gothenburg, Gothenburg, Sweden, 4 Department of Infectious Diseases, Sahlgrenska University Hospital, Gothenburg, Region Västra Götaland, Sweden

☯ These authors contributed equally to this work.
* isabel.vigmo@gu.se

## Abstract

### Background

B cell abnormalities are an early feature of human immunodeficiency virus-1 (HIV) infection, leading to hypergammaglobulinemia.

### Objective(s)

The aim of this study was to evaluate serum levels of immunoglobulin G (IgG) as well as the inflammatory biomarkers neopterin, β2-microglobulin and albumin in a large cohort of untreated as well as virologically suppressed people with HIV (PWH) on antiretroviral therapy (ART).

### Methods

We conducted a cross-sectional analysis of blood samples from untreated PWH and virologically suppressed participants on ART. Of 750 collected blood samples, 267 were from ART-suppressed participants and 483 from untreated participants. Levels of IgG, HIV-RNA, CD4+ T cells, β2-microglobulin, neopterin and albumin were analyzed with regard to age, sex and stage of HIV infection.

### Results

Hypergammaglobulinemia was present in all subgroups of untreated PWH except in those with primary HIV infection. Participants with a CD4+ T cell count between 50 and 349 cells/μL exhibited the highest IgG concentrations. Elevated IgG concentrations prevailed in 24% of participants on ART. Serum neopterin and β2-microglobulin levels were significantly elevated in untreated PWH. They decreased in participants on ART, but remained abnormally elevated in approximately half of those.

**Data availability statement:** Due to the sensitivity of the data behind this study's findings and restrictions related to participant confidentiality, data access is restricted. Upon reasonable request, data access can be granted from the Swedish National Data Service. URL: https://doi.org/10.5878/v1kj-fq44.

**Funding:** This work was supported by the Swedish state under an agreement between the Swedish government and the county councils (the ALF agreement; https://www.alfvastragotaland.se/; ALFGBG-942707 (JR) and ALFGBG-965885 and 1005848 (MG)). The funders had no role in study design, data collection and analysis, decision to publish, or preparation of the manuscript.

**Competing interests:** MG has received research grants from Gilead Sciences and honoraria as speaker, DSMB committee member and/or scientific advisor from Amgen, AstraZeneca, Biogen, Bristol-Myers Squibb, Gilead Sciences, GlaxoSmithKline/ViiV, Janssen-Cilag, MSD, Novocure, Novo Nordic, Pfizer and Sanofi (outside of submitted work). All mentioned engagements have concluded and are not ongoing. The other authors report there are no competing interests to declare.

## Conclusion

Although IgG levels were normalized in a majority of participants on ART, hypergammaglobulinemia prevailed in about a quarter of cases, indicating a remaining B cell hyperactivity despite viral suppression. The finding of elevated levels of inflammatory biomarkers despite ART suppression is suggestive of residual inflammatory activity and parallels the finding of persisting hypergammaglobulinemia.

## Introduction

Human immunodeficiency virus-1 (hereafter HIV) infection is characterized by a profound impairment of both the cellular and the humoral arms of the immune system. The immunological aberrancies in the B cell compartment caused by HIV infection include a polyclonal B cell activation and subsequent hypergammaglobulinemia [1,2]. Increased serum immunoglobulin G (IgG) levels up to twice the normal upper level have previously been reported in people with HIV (PWH) [2]. This hypergammaglobulinemia is an early sign of HIV infection that may precede quantitative deficiencies of CD4$^+$ T cells, and comprises both HIV-specific, as well as non-specific antibodies [3–5].

Despite elevated immunoglobulin levels, PWH exhibit a deficit in functional antibody response to vaccines designed to elicit a humoral response, highlighting a dysfunctionality of the B cell compartment [6,7]. The B cell dysfunction seen in HIV infection has been attributed to various mechanisms including dysregulated cytokine production [8,9], direct impact of viral replication, and a deficiency in T cell assistance of B cell maturation, related to an HIV-induced T cell defect [10]. However, the exact mechanisms responsible for inducing the excessive immunoglobulin production in HIV infection remain incompletely understood.

Antiretroviral therapy (ART) leads to suppression of HIV-RNA levels and restores the functions of the immune system. The reconstitution of T cell immunity with effective ART has been extensively documented [11–13]. There is, however, limited data on functional B cell reconstitution as represented by total IgG concentrations. The initiation of ART and subsequent viral suppression causes a reduction in the numbers of IgG-producing B cells, as well as a decline of HIV-specific antibody titres [4,14]. However, the majority of the circulating IgG in hypergammaglobulinemic PWH has been shown to comprise non-specific antibodies induced by polyclonal immune activation and not a virus-specific humoral response [5].

A complete normalization of hypergammaglobulinemia in PWH receiving effective ART has previously been reported [14]. Other studies have shown a decline in serum IgG concentrations among PWH receiving ART, however without normalized levels in nearly half of the individuals even when viral load was undetectable [15,16].

Current data are limited in depicting the correlation between IgG levels, immunological status, and levels of plasma viremia in PWH. Moreover, data on IgG levels in virologically controlled patients on modern ART are largely missing. The aim of the present study was to evaluate IgG levels in a comprehensive cohort of PWH,

including untreated individuals with varying CD4$^+$ T cell counts and HIV-RNA levels, as well as those with viral suppression during ART. Additionally, we investigated the association between serum levels of IgG, immune activation biomarkers (neopterin and β2-microglobulin), and albumin at different stages of HIV infection, considering the participants' age and sex.

## Methods

### Study design and participants

This is a cross-sectional study of blood samples ($n$ = 821) collected from a total of 584 adult PWH (age ≥ 18 years) and 71 seronegative controls. The blood samples were collected in a standardized manner during outpatient clinic visits as part of the Gothenburg HIV CSF Study Cohort at the Department of Infectious Diseases, Sahlgrenska University Hospital, Gothenburg, Sweden, ongoing since 1985 [17]. A total of 750 blood samples were collected from 584 PWH. Of these, 267 were taken in individuals with ongoing ART and 483 from those without ongoing treatment. 166 participants provided blood samples twice at different time points and were hence included in both treated and untreated groups. Data from the Gothenburg HIV CSF Study Cohort and the HIV-negative control group were accessed for research purposes on 24/04/2023 from a prospectively collected material. Data became available at the time of sample collection and were compiled for analysis between 13/11/1985 and 14/03/2023. Data on hepatitis C status was extracted from the Swedish InfCareHIV cohort [18].

### Definitions

The samples from the HIV-positive participants were categorized according to eight predefined categories representing immunologically different stages of HIV infection as previously described [19,20]: primary HIV infection (PHI; within 12 months of initial HIV infection), five groups characterized by CD4$^+$ T cell counts (≥ 500; 350 − 499; 200 − 349, 50 − 199 and < 50 cells/µL), ongoing infection or malignancy (OIM), and those on ART. Five study participants without detectable virus in blood (HIV-RNA < 50 copies/mL) in the absence of ART, i.e., elite controllers, were included in study groups defined by CD4$^+$ T cell count. In line with previous studies, the diagnosis of PHI was based on a combination of clinical histories and documented seroconversion, nucleic acid testing or enzyme-linked immunosorbent assay (ELISA) testing [21]. OIM was defined as the presence of an opportunistic or non-opportunistic bacterial, viral or parasitic infection or active cancer in association with or within two weeks of blood sampling. ART suppression was defined as ongoing ART with plasma HIV-RNA < 50 copies/mL for ≥ 6 months and included antiretroviral regimens with two or more active agents (non-nucleoside reverse transcriptase inhibitors, nucleoside reverse transcriptase inhibitors, integrase inhibitors and/or protease inhibitors) according to guidelines existing at the time of inclusion [22–27]. Among the HIV-positive participants, only those in the ART group received antiretroviral therapy. Participants meeting the criteria for the categories PHI, OIM and ART were designated to these irrespective of CD4$^+$ T cell counts. The 166 participants who provided two samples were included once in the ART group and once in an untreated group. Accordingly, no participant contributed more than a single sample to any study group.

The HIV-negative control group ($n$ = 71) consisted of 44 individuals receiving HIV preexposure prophylaxis (PrEP) with tenofovir disoproxil fumarate (TDF)/emtricitabine (FTC) and 27 volunteers (healthcare workers, students and their relatives) without PrEP [28].

### Participant consent

This study involved participants from the Gothenburg HIV CSF Study Cohort [17] approved by the Regional Ethics Review Board in Gothenburg, Sweden (reference number Ö588 − 01). The participation of the HIV-negative control group was granted by the Regional Ethics Review Board in Gothenburg, Sweden (Dnr: 060 − 18). The study was conducted in

accordance with the ethical principles set out in the declaration of Helsinki, and all participants provided written informed consent prior to their participation in the study. Information that could identify individual participants was available to the authors during and after data collection.

## Laboratory methods

All blood samples were obtained on the same occasion for each patient and analyzed consecutively according to current clinical routines.

Serum IgG and albumin concentrations were determined with nephelometry (Behring Nephelometer Analyser, Behringwerke AG, Marburg, Germany) or by immunoturbidimetry on a Cobas instrument (Roche Diagnostics, Penzberg, Germany). The reference interval for IgG was 6.8−15.0 g/L. Hypergammaglobulinemia was defined as an IgG concentration above 15 g/L. Based on the laboratory's validation and external quality control data, there were no clinically relevant differences in measured IgG concentrations between nephelometry and immunoturbidimetry. Accordingly, results across the study period are comparable over time. The reference interval for albumin was age dependent: 36−48 g/L for individuals aged 18 to < 41 years, 36−45 g/L for those aged 41 to < 70 years and 34−45 g/L for individuals aged 70 years or older.

Serum neopterin concentrations were analyzed using a commercially available immunoassay (NEOPT-SCR.EIA 384 Det., Thermo Fisher Scientific – BRAHMS GmbH, Henningsdorf, Germany) with an upper normal reference level of 9.1 nmol/L [29].

Levels of serum β2-microglobulin were measured using the N Latex β2M kit on the Atellica NEPH 630 System (Siemens Healthcare GmbH, Erlangen, Germany), with upper normal reference levels below 1.8 mg/L for individuals under 50 years old and below 2.1 mg/L for those aged 50 years or older.

Plasma HIV-RNA levels were assayed by the Roche Amplicor Monitor version 1.5 (Roche Molecular Systems, Brandenburg, NJ) or Roche Taqman assay, version 1 or 2 (Hoffman La-Roche, Basel, Switzerland). Peripheral blood CD4+ T cell counts were performed in the local clinical laboratory using routine methods.

## Statistical methods

All calculations were two-sided with alpha 0.05 and performed in Graph Pad Prism (version 10.0.2, GraphPad Software, San Diego, CA) or SPSS (IBM SPSS version 29). Variables were reported as median (interquartile range (IQR)) unless otherwise stated. All participants with an HIV-RNA below 20 copies per mL were assigned a value of 10 copies per mL in the statistical analysis. HIV-RNA concentrations were $log_{10}$-transformed. Each HIV-positive group was compared to the HIV-negative control group using the Mann−Whitney U test for continuous variables. The Chi-square test was used to compare the proportion of participant samples with IgG levels above 15 g/L between the ART group and the HIV-negative control group. Partial correlation controlling for age, was performed between IgG and HIV-RNA, CD4+ T cells, neopterin, β2-microglobulin and albumin. The correlation coefficient between age and IgG was determined using the Spearman rank correlation test. For Table 2, we applied Holm-Bonferroni correction. Multivariable analysis adjusting for age and CD4+ T cell count was performed with log-transformed IgG levels as the dependent variable, comparing HIV-negative controls with the ART group and an untreated group consisting of the five groups stratified by CD4+ T cell count.

## Results

Median age of PWH was 37 (IQR 31−47) and 48 (IQR 39−57) years for untreated and ART-suppressed participants, respectively. Women constituted 33% (245/750) of the HIV-positive cohort. The HIV-negative control group was younger (median age 35 (IQR 29−43) years) with 15% women (11/71) (Table 1). There were no women among the HIV-negative controls with PrEP. Background clinical and laboratory data pertaining the different study groups are presented in Table 1. The most common infections in the OIM group included pneumocystis jirovecii pneumonia, tuberculosis, syphilis, candida

**Table 1. Participant characteristics at sampling.**

| Groups | n | Age, years, median (IQR) | Female sex, n (%) | Plasma HIV-RNA, $\log_{10}$ copies/mL, median (IQR) [a] | CD4$^+$T cells/µL, median, (IQR) [b] |
|---|---|---|---|---|---|
| **HIV-negative controls** | 71 | 35 (29-43) | 11 (15.5) | – | 830 (633-1100) |
| • With PrEP | 45 | 35 (29-42) | 0 (0) | – | 830 (620-1025) |
| • Without PrEP | 26 | 34 (27-42) | 11 (42.3) | – | 835 (685-1100) |
| **PHI** | 45 | 41 (28-47) | 4 (8.90) | 5.17 (4.0-6.20) | 450 (300-605) |
| **CD4$^+$ ≥ 500** | 59 | 35 (29-47) | 23 (39.0) | 3.84 (3.37-4.42) | 632 (563-795) |
| **CD4$^+$ 350–499** | 50 | 37 (29-45) | 18 (36.0) | 4.20 (3.73-4.64) | 408 (390-458) |
| **CD4$^+$ 200–349** | 90 | 35 (29-43) | 37 (41.1) | 4.69 (4.20-5.13) | 252 (226-300) |
| **CD4$^+$ 50–199** | 81 | 38 (32-49) | 30 (37.0) | 5.02 (4.44-5.54) | 130 (90-168) |
| **CD4$^+$ < 50** | 29 | 36 (30-42) | 5 (17.2) | 5.36 (4.84-5.72) | 20 (10-38) |
| **OIM** | 129 | 41 (35-49) | 37 (28.7) | 5.25 (4.64-5.78) | 53 (20-205) |
| **ART** | 267 | 48 (39-57) | 91 (34.1) | < 1.70 | 610 (410-800) |

IQR, interquartile range; HIV, human immunodeficiency virus-1; PrEP, pre-exposure prophylaxis, PHI, primary HIV-1 infection; OIM, ongoing infection or malignancy; ART, antiretroviral therapy with plasma-HIV-RNA<50 copies/mL for ≥ 6 months. The table shows 821 participant samples from 584 HIV-positive participants and 71 HIV-negative controls. 166 HIV-positive participants were included twice, once in an untreated group and once in the ART group.

[a]Missing data by random, n = 21 study participant samples (2.8%).

[b]Missing data by random, n = 4 study participant samples (0.49%).

esophagitis, and herpes zoster (S1 Table). Participants in the ART group had been virally suppressed for a median of 58 (23 − 111) months at the time of sampling.

## IgG concentrations

IgG concentrations were significantly elevated in all categories of PWH regardless of CD4$^+$ T cell count when compared with HIV-negative controls. The highest IgG concentrations were found in participants with a CD4$^+$ T cell count between 50 and 349 cells/µL, who had more than a twice as high IgG level compared to the controls (Table 2, Fig 1). In total, 141/171 (82%) of participants with a CD4$^+$ T cell count between 50–349 cells/µL had IgG concentrations above the upper reference level of 15 g/L. Participants with CD4$^+$ T cells < 50/µL exhibited slightly lower median IgG concentrations with hypergammaglobulinemia present in 20/29 (69%) of these participants. Despite high levels of HIV-RNA, the PHI participants had IgG levels within the normal range. No significant difference was seen in IgG levels when comparing HIV-negative participants with and without PrEP (10.0 vs 11.0 g/L, P = 0.861) (Table 2). The results for the control group comparisons remained the same when stratifying for the HIV-negative control groups with and without PrEP. In a multivariable analysis with log-transformed IgG levels, adjusting for sex and log CD4$^+$ T cell count, the untreated group (only the five groups stratified by CD4$^+$ T cell count) showed IgG levels that were two-fold higher than those of HIV-negative controls. The ART group had 13% higher IgG levels than HIV-negative controls. Both comparisons were highly significant (P < 0.001).

Median IgG concentration among participants on ART was within the normal reference range, 12.0 (10.0 − 15.0) g/L. Nevertheless, 65/267 (24%) of the ART participants exhibited IgG concentrations above the upper reference level compared to 1/71 (1.4%) of the HIV-negative controls (P < 0.001). Nadir CD4$^+$ T cell counts were available for 90% (241/267) of the ART participants (S1 Fig). There was a weak negative correlation between CD4$^+$ T cell nadir and IgG concentrations in the ART-suppressed group (S2 Fig).

Both untreated and ART-suppressed women with HIV exhibited higher IgG levels than men in corresponding groups (22.5 vs. 19.0 g/L, P < 0.001 and 14.0 vs. 11.5 g/L, P < 0.001, respectively) (Fig 2). This gender difference was consistent throughout, although statistical significance was not reached in all untreated HIV-positive study groups (data not shown).

**Table 2. Serum IgG and inflammatory biomarkers in samples from HIV-positive participants and negative controls.**

|  |  | Median values (IQR) |  |  |  |
|---|---|---|---|---|---|
| Groups | n | S IgG (g/L) | S neopterin (nmol/L) [a] | S β2-microglobulin (mg/L) [b] | S albumin (g/L) |
| **HIV-negative controls** | 71 | 10.0 (9.20-12.0) | 7.60 (6.20-10.7) | 1.80 (1.60-2.10) | 43.0 (41.0-45.0) |
| • With PrEP | 45 | 10.0 (9.30-12.0) | 8.90 (7.40-11.9) | 1.90 (1.70-2.20) | 43.0 (40.0-45.0) |
| • Without PrEP | 26 | 11.0 (8.73-12.0) | 6.75 (5.53-7.25) | 1.60 (1.50-1.70) | 43.5 (42.0-45.8) |
| **PHI** | 45 | 13.0 (11.0-15.0) *** | 16.5 (9.75-23.0) *** | 2.85 (2.20-3-40) *** | 42.0 (38.0-46.0) |
| **CD4 ≥ 500** | 59 | 18.0 (14.0-23.0) *** | 9.50 (8.25-11.9) *** | 2.40 (2.00-2.80) *** | 42.0 (40.0-46.0) |
| **CD4 350–499** | 50 | 21.5 (17.3-27.0) *** | 10.4 (8.40-17.2) *** | 2.50 (2.00-3.20) *** | 40.5 (38.0-43.0) *** |
| **CD4 200–349** | 90 | 22.5 (17.0-29.0) *** | 16.7 (9.80-22.6) *** | 3.00 (2.40-3.80) *** | 40.0 (37.3-43.8) *** |
| **CD4 50–199** | 81 | 23.0 (19.0-29.0) *** | 23.0 (15.0-39.3) *** | 3.75 (2.93-4.98) *** | 38.0 (35.0-42.0) *** |
| **CD4 < 50** | 29 | 20.0 (15.0-24.0) *** | 24.0 (19.8-32.8) *** | 3.80 (3.10-4.60) *** | 39.0 (34.0-42.0) *** |
| **OIM** | 129 | 20.0 (15.0-27.0) *** | 33.5 (23.1-49.7) *** | 4.20 (3.60-5.28) *** | 33.0 (28.0-37.0) *** |
| **ART** | 267 | 12.0 (10.0-15.0) *** | 8.30 (5.95-11.7) | 2.00 (1.70-2.30) ** | 42.0 (39.0-44.0) *** |

IgG, immunoglobulin G; HIV, human immunodeficiency virus-1; PHI, primary HIV-1 infection; OIM, ongoing infection or malignancy; ART, antiretroviral therapy with plasma-HIV-RNA < 50 copies/mL for ≥ 6 months. Reference intervals: IgG (6.8 − 15.0 g/L); neopterin (≤ 9.1 nmol/L); β2-microglobulin (< 1.8 mg/L (< 50 years); < 2.1 mg/L (≥ 50 years)); albumin (36 − 48 g/L (18 to < 41 years); 36 − 45 g/L (41 to < 70 years); 34 − 45 g/L (≥ 70 years)). *** $P < 0.001$; ** $P < 0.01$ (still significant after Holm-Bonferroni correction of the 32 tests). The table shows 821 samples from 584 HIV-positive participants and 71 HIV-negative controls. 166 HIV-positive participants were included twice, once in an untreated group and once in the ART group. All comparisons between the respective HIV-positive groups were vs all HIV-negative controls (Mann−Whitney U test).

[a]Missing data by random, $n = 54$ study participant samples (6.6%). Excluded in analysis.

[b]Missing data by random, $n = 36$ study participant samples (4.4%). Excluded in analysis.

The opposite was seen among the HIV-negative controls without PrEP, although the difference was not statistically significant.

Active hepatitis C was found in 23 of 750 samples (3.1%) from PWH, while data was missing for 120 samples (16%). All HIV-negative participants tested negative for hepatitis C. Sensitivity analysis, performed after excluding participants with active or unknown hepatitis C status, showed no change in the finding of significantly higher IgG levels in the respective HIV-positive groups (data not shown).

## Inflammatory biomarkers

The highest concentrations of serum neopterin and β2-microglobulin were found in participants with ongoing infection or malignancy and the lowest among participants on ART (Table 2, Fig 1). Among ART-suppressed participants, 39% (90/231) had elevated concentrations of neopterin and 58.1% (155/267) had elevated β2-microglobulin levels. HIV-negative participants with PrEP showed higher levels of neopterin and β2-microglobulin than HIV-negative participants without PrEP (8.9 vs 6.75 nmol/L, $P < 0.0001$ and 1.9 vs 1.6 mg/L, $P < 0.001$, respectively) (Table 2), as reported in our previous work [28,30].

The lowest serum albumin concentrations were seen among participants with ongoing infection or malignancy (Table 2, Fig 1). Albumin concentrations were within the normal range in progressive stages of HIV infection with CD4+ T cell depletion down to 50 cells/μL. The vast majority of participants on ART (250/267; 93.6%) had normal albumin concentrations. There was no significant difference in albumin levels in HIV-negative participants with respective without PrEP (43.0 vs 43.5 g/L, $P = 0.297$) (Table 2).

## Correlations between IgG and immune status, inflammatory biomarkers and age

No significant correlation was found between IgG and HIV-RNA (Fig 3A) or neopterin (Fig 3C). There was a weak negative correlation between IgG concentrations and CD4+ T cells (Fig 3B) and a weak positive correlation between IgG and

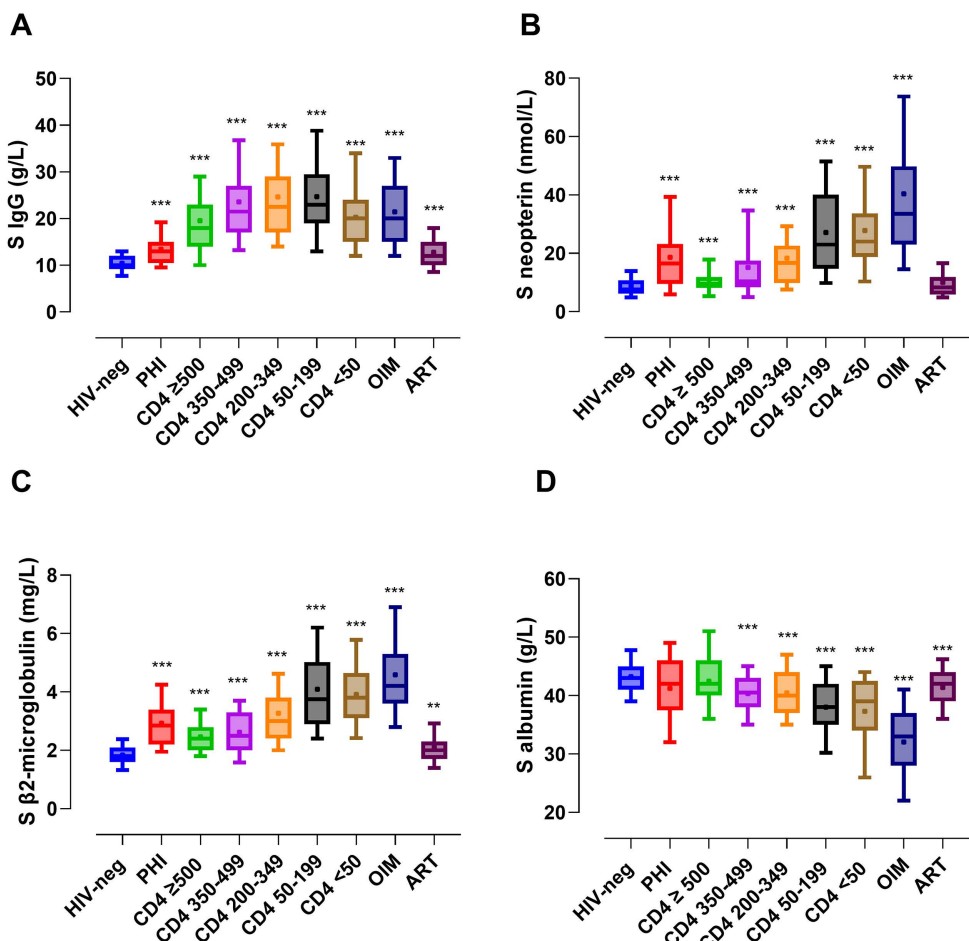

**Fig 1. Serum IgG and inflammatory biomarkers in different study groups.** IgG, immunoglobulin G; HIV, human immunodeficiency virus-1; PHI, primary HIV infection; OIM, ongoing infection or malignancy; ART, antiretroviral treatment. The HIV-negative group includes participants with and without PrEP. Boxes represent interquartile range with median (line) and mean (+). Whiskers depict the 10th and 90th percentiles. Reference intervals: IgG (6.8 − 15.0 g/L); neopterin (≤ 9.1 nmol/L); β2-microglobulin (< 1.8 mg/L (< 50 years); < 2.1 mg/L (≥ 50 years)); albumin (36 − 48 g/L (18 to < 41 years); 36 − 45 g/L (41 to < 70 years); 34 − 45 g/L (≥ 70 years)). All statistical comparisons are performed versus all HIV-negative controls (Mann−Whitney U test). *** $P < 0.001$; ** $P < 0.01$.

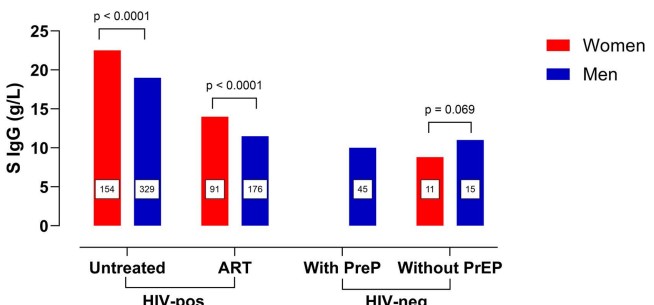

**Fig 2. Median serum IgG concentrations in men and women.** IgG, immunoglobulin G; HIV, human immunodeficiency virus-1. Samples from treated ($n = 267$) and untreated ($n = 483$) HIV-positive participants and HIV-negative controls ($n = 71$), the latter subdivided into participants with ($n = 45$) and without ($n = 26$) pre-exposure prophylaxis (PrEP). IgG reference interval: 6.8 − 15.0 g/L. There were no women among the HIV-negative controls with PrEP. The number of participant samples is enclosed in the bars.

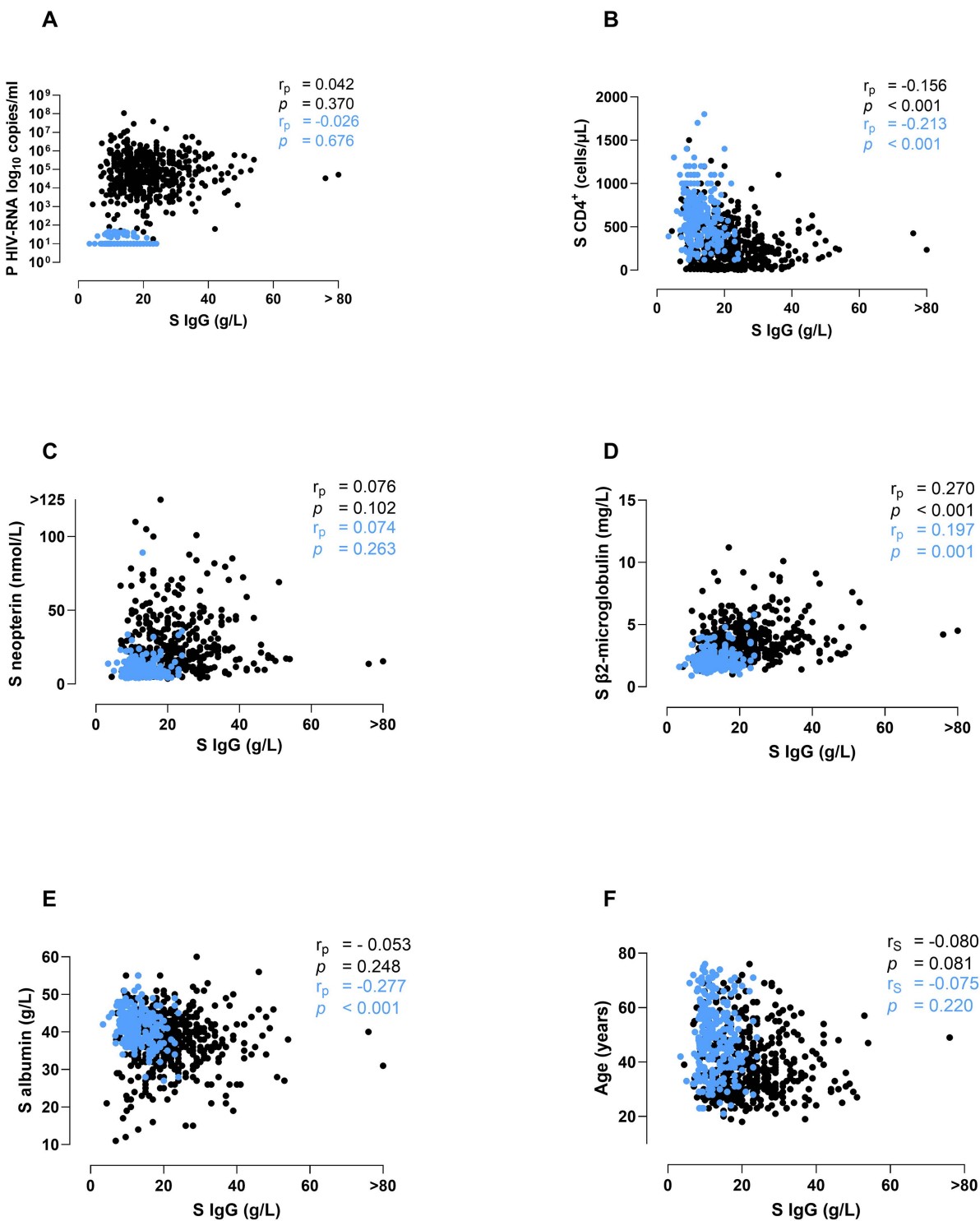

**Fig 3. Correlations between serum IgG concentrations and HIV-RNA levels, CD4+ T cell count, inflammatory biomarkers and age.** (A) Plasma HIV-RNA ($\log_{10}$ copies/mL), (B) CD4+ T cell count (cells/µL), (C) serum neopterin (nmol/L), (D) serum β2-microglobulin (mg/L), (E) serum albumin (g/L), (F) age (years). Correlation coefficients were explored with partial correlation for age ($t_p$; A-E) or using the Spearman rank correlation test (t; F). Samples from participants on ART are indicated in blue, samples from participants without ART in black. IgG, immunoglobulin G. Reference intervals: IgG (6.8 − 15.0 g/L); neopterin (≤ 9.1 nmol/L); β2-microglobulin (< 1.8 mg/L (< 50 years); < 2.1 mg/L (≥ 50 years)); albumin (36 − 48 g/L (18 to < 41 years); 36 − 45 g/L (41 to < 70 years); 34 − 45 g/L (≥ 70 years)).

β2-microglobulin (Fig 3D) in both treated and untreated participants. A weak positive correlation was displayed between albumin and IgG in ART-suppressed participants (Fig 3E). Further, age did not correlate significantly with IgG levels in neither untreated nor ART-suppressed participants (Fig 3F).

## Discussion

In this study, we investigated IgG levels and inflammatory biomarkers in PWH with and without ART. Hypergammaglobulinemia was observed across the spectrum of CD4+ T cell count in untreated PWH, except in participants with PHI. It was more pronounced in women and did not correlate with age. Among ART-suppressed participants, elevated levels of IgG and inflammatory biomarkers were present in approximately one-quarter and one-half, respectively.

The highest IgG concentrations were seen among individuals with CD4+ T cell strata of 50−199 and 200−349 cells/µL, thus in participants with a more advanced HIV infection with a high viral burden and CD4+ T cell depletion. A less prominent hypergammaglobulinemia was found in participants with the most pronounced immunosuppression (CD4+ T cells < 50/µL). However, the small sample size of this group might have influenced the results. In fact, all groups with a CD4+ T cell count <=500 cells/µL had a median IgG concentration at least twice as high as in the HIV-negative control group. IgG levels were significantly elevated in participants with PHI when compared to HIV-negative controls, although still within the normal range. A previous study has demonstrated a high frequency of IgG-producing B cells, without corresponding hypergammaglobulinemia in participants with PHI [4]. This indicates that the activation of the B cell compartment and the subsequent development of hypergammaglobulinemia is a process that takes considerable time.

Although the majority of virally suppressed participants on ART in the present study exhibited total IgG concentrations within the normal range, hypergammaglobulinemia was still present in about a quarter (24%), indicating a remaining B cell hyperactivity. This is in agreement with two previous small studies that also showed persistent hypergammaglobulinemia in 45% and 37% in ART-suppressed PWH, respectively [15,16].

In order to investigate the potential impact of preceding immunodeficiency, we evaluated CD4+ T cell nadir in the ART participants, but there was only a weak correlation with IgG levels. This has also been demonstrated in previous studies [15,16], and indicates that hypergammaglobulinemia may arise and persist also when ART is initiated before a significant CD4+ T cell depletion has occurred.

It has been shown that changes in total IgG concentrations follow the fluctuations in HIV-RNA levels in participants with treatment failure, implying a potential dependency on viral replication [14]. The time frame needed for normalization of HIV-induced hypergammaglobulinemia after viral suppression is reached has not been established, but was achieved at follow-up after 52−84 weeks of ART in previous studies [14,15]. Given the number of participants with persisting hypergammaglobulinemia despite viral suppression in our study, it is unlikely that HIV replication alone can fully explain the presence of elevated IgG concentrations. Further, the IgG levels of participants on ART in this study were analyzed after six months, but often longer, of viral suppression and it may be of doubt that the persistently elevated IgG levels would resolve in a longer-term follow-up.

Our study cohort displayed higher IgG concentrations in women than in men, which agrees with a previous study of PWH [31]. A large meta-analysis revealed lower IgG levels in men than in women. While not significant in a random effects model, it was highly significant in a fixed-effects model [32]. Although there was no significant difference in IgG levels among HIV-negative participants with and without PrEP, it should be noted that the participants on PrEP were exclusively men. This leaves IgG levels in women on PrEP unexplored. Among HIV-negative participants without PrEP, women had lower IgG levels than men. Nevertheless, women only comprised a small part of this study group, and this difference was not statistically significant. The male predominance of this study reflects the fact that a large proportion of the samples were obtained early in the HIV pandemic when HIV was mainly diagnosed among men who have sex with men in Sweden.

Hepatitis C has a common mode of transmission as HIV and may also cause elevated IgG concentrations [33]. We therefore investigated the presence of co-infection with hepatitis C but only identified active hepatitis C in a small percentage of the participants. Excluding individuals with active or unknown hepatitis C status did not alter the main findings, supporting that the observed hypergammaglobulinemia is primarily HIV-driven.

Generalized immune activation is a well-known feature of HIV infection, and multiple inflammatory biomarkers remain elevated also when the viral replication is suppressed by ART [34]. Increased levels of neopterin and β2-microglobulin reflect mainly macrophage and lymphocyte activation, respectively [19,35]. The PWH in the present study had elevated concentrations of serum neopterin and β2-microglobulin that increased with declining CD4+ T cell counts, which agrees with previous studies [36,37]. Even though participants on ART had median levels of neopterin and β2-microglobulin within the normal range, approximately half had persisting elevated concentrations. Thus, our finding is suggestive of residual inflammatory activity despite ART suppression and parallels the persistent hypergammaglobulinemia observed in this group. Two previous studies from our group have shown that HIV-negative people on PrEP have higher serum levels of neopterin and β2-microglobulin compared to HIV-negative controls not on PrEP [28,30]. The exact mechanism remains unclear, but both lifestyle factors and co-infections have been proposed as contributing factors. Serum albumin levels below the lower reference level were only found among participants with ongoing infection or malignancy and could be regarded as a general disease marker associated with acute concomitant illness.

Immunosenescence affects the B cell compartment with declining numbers of naïve and peripheral B cells, as well as a reduced B cell diversity [38–40]. However, we found no significant correlation between IgG and age in PWH. This aligns with a large meta-analysis in which no association between IgG and age in the general population was seen [32]. It should be noted that the untreated and ART-suppressed PWH in this study population had a median age of 37 and 48 years, respectively. Thus, the ability to drive B cell hyperactivation with consequent hypergammaglobulinemia is insufficiently characterized in older PWH.

Persistently elevated IgG concentrations despite ART suppression as a manifestation of chronic B cell activation and/or dysfunction may be of clinical importance. It is likely that B cell hyperactivation plays a role in the increased frequency of B cell neoplasms seen in PWH [41,42]. The incidence of B cell lymphomas, *e.g.*, non-Hodgkin lymphoma has decreased significantly but is still elevated in PWH compared to the general population despite the widespread implementation of ART [43]. Moreover, the risk of invasive pneumococcal disease is reduced during ART suppression but is still significantly higher compared to the general population [44], which could indicate a persisting dysfunctional humoral immune response.

To our knowledge, this is the largest study characterizing IgG levels in a cohort of untreated and ART-suppressed PWH. The number of participants is a main strength of the study, as well as the comprehensive study period (1985 – 2023) covering samples from the pre-ART era to the era of modern treatment. Limitations include the lack of longitudinal follow-up of IgG levels in the study participants as well as the absence of women in the HIV-negative participants on PrEP, which may influence the interpretation of between-group comparisons. Additionally, data on other chronic co-infections (e.g., Epstein–Barr virus, syphilis, and hepatitis B) were not available, which is another limitation of the study.

To conclude, this study showed increased IgG concentrations in all groups with PWH except participants with PHI. Although participants on ART had significantly lower levels of IgG compared to untreated PWH excluding PHI, still one fourth exhibited hypergammaglobulinemia. This indicates that viral replication is not the sole driving factor behind B cell activation in PWH. The finding of a remaining immune activation was reinforced by elevated inflammatory biomarkers (β2-microglobulin, neopterin). Although serum IgG levels are not routinely measured in clinical practice, the analysis is readily available and may indicate the degree of B cell dysfunction and reconstitution in PWH.

## Supporting information

**S1 Table. Ongoing infection or malignancy.** Ongoing infection or malignancy was defined as the presence of an opportunistic or non-opportunistic bacterial, viral or parasitic infection or active cancer in association with or within two weeks of blood sampling. *n*, number of diagnoses; CMV, cytomegalovirus; PML, progressive multifocal leukoencephalopathy; VZV, varicella- zoster virus. Nine out of 129 participants were diagnosed with more than one ongoing infection.
(DOCX)

**S1 Fig. Distribution of CD4$^+$ T cell nadir (cells/µL) in participants on ART (*n* = 241).** ART; antiretroviral therapy with plasma HIV-RNA < 50 copies/mL for ≥ 6 months. Data on CD4$^+$ T cell nadir was missing for 26 out of 267 participants (9.7%).
(TIF)

**S2 Fig. Correlation between serum IgG and CD4$^+$ T cell nadir in participants on ART (*n* = 241).** IgG, immunoglobulin G; ART; antiretroviral therapy with plasma HIV-RNA < 50 copies/mL for ≥ 6 months. Data on CD4$^+$ T cell nadir was missing for 26 out of 267 participants (9.7%). IgG reference interval: 6.8 − 15.0 g/L.
(TIF)

## Author contributions

**Conceptualization:** Magnus Gisslén.

**Data curation:** Magnus Gisslén.

**Formal analysis:** Isabel Vigmo, Staffan Nilsson.

**Funding acquisition:** Magnus Gisslén.

**Methodology:** Staffan Nilsson, Magnus Gisslén, Johanna Karlsson.

**Supervision:** Magnus Gisslén, Josefina Robertson, Johanna Karlsson.

**Writing – original draft:** Isabel Vigmo.

**Writing – review & editing:** Aylin Yilmaz, Magnus Gisslén, Josefina Robertson, Johanna Karlsson.

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
