## [Decision Letter · Decision Letter 0]

10 Sep 2025

PONE-D-25-36523Hypergammaglobulinemia in treated and untreated people with HIVPLOS ONE

Dear Dr. Vigmo,

Thank you for submitting your manuscript to PLOS ONE. After careful consideration, we feel that it has merit but does not fully meet PLOS ONE’s publication criteria as it currently stands. Therefore, we invite you to submit a revised version of the manuscript that addresses the points raised during the review process.

We look forward to receiving your revised manuscript.

Kind regards,

Sepiso K. Masenga, PhD

Academic Editor

PLOS ONE

**Journal Requirements:**

1. When submitting your revision, we need you to address these additional requirements. Please ensure that your manuscript meets PLOS ONE's style requirements, including those for file naming. The PLOS ONE style templates can be found at https://journals.plos.org/plosone/s/file?id=wjVg/PLOSOne_formatting_sample_main_body.pdf and https://journals.plos.org/plosone/s/file?id=ba62/PLOSOne_formatting_sample_title_authors_affiliations.pdf 2. Thank you for stating in your Funding Statement: This work was supported by the Swedish state under an agreement between the Swedish government and the county councils (the ALF agreement; https://www.alfvastragotaland.se/; ALFGBG-942707 (JR) and ALFGBG-965885 and 1005848 (MG)). The funders had no role in study design, data collection and analysis, decision to publish, or preparation of the manuscript  Please provide an amended statement that declares *all* the funding or sources of support (whether external or internal to your organization) received during this study, as detailed online in our guide for authors at http://journals.plos.org/plosone/s/submit-now. Please also include the statement “There was no additional external funding received for this study.” in your updated Funding Statement. Please include your amended Funding Statement within your cover letter. We will change the online submission form on your behalf. 3. Thank you for stating the following in the Competing Interests section: MG has received research grants from Gilead Sciences and honoraria as speaker, DSMB committee member and/or scientific advisor from Amgen, AstraZeneca, Biogen, Bristol-Myers Squibb, Gilead Sciences, GlaxoSmithKline/ViiV, Janssen-Cilag, MSD, Novocure, Novo Nordic, Pfizer and Sanofi (outside of submitted work). All mentioned engagements have concluded and are not ongoing. The other authors report there are no competing interests to declare. We note that one or more of the authors are employed by a commercial company.  a. Please provide an amended Funding Statement declaring this commercial affiliation, as well as a statement regarding the Role of Funders in your study. If the funding organization did not play a role in the study design, data collection and analysis, decision to publish, or preparation of the manuscript and only provided financial support in the form of authors' salaries and/or research materials, please review your statements relating to the author contributions, and ensure you have specifically and accurately indicated the role(s) that these authors had in your study. You can update author roles in the Author Contributions section of the online submission form. Please also include the following statement within your amended Funding Statement. “The funder provided support in the form of salaries for authors, but did not have any additional role in the study design, data collection and analysis, decision to publish, or preparation of the manuscript. The specific roles of these authors are articulated in the ‘author contributions’ section.”If your commercial affiliation did play a role in your study, please state and explain this role within your updated Funding Statement.  b. Please also provide an updated Competing Interests Statement declaring this commercial affiliation along with any other relevant declarations relating to employment, consultancy, patents, products in development, or marketed products, etc.   Within your Competing Interests Statement, please confirm that this commercial affiliation does not alter your adherence to all PLOS ONE policies on sharing data and materials by including the following statement: "This does not alter our adherence to  PLOS ONE policies on sharing data and materials.” (as detailed online in our guide for authors http://journals.plos.org/plosone/s/competing-interests) . If this adherence statement is not accurate and  there are restrictions on sharing of data and/or materials, please state these. Please note that we cannot proceed with consideration of your article until this information has been declared. Please include both an updated Funding Statement and Competing Interests Statement in your cover letter. We will change the online submission form on your behalf. 4. Please note that your Data Availability Statement is currently missing the DOI/accession number of each dataset OR a direct link to access each database. If your manuscript is accepted for publication, you will be asked to provide these details on a very short timeline. We therefore suggest that you provide this information now, though we will not hold up the peer review process if you are unable. 5. When completing the data availability statement of the submission form, you indicated that you will make your data available on acceptance. We strongly recommend all authors decide on a data sharing plan before acceptance, as the process can be lengthy and hold up publication timelines. Please note that, though access restrictions are acceptable now, your entire data will need to be made freely accessible if your manuscript is accepted for publication. This policy applies to all data except where public deposition would breach compliance with the protocol approved by your research ethics board. If you are unable to adhere to our open data policy, please kindly revise your statement to explain your reasoning and we will seek the editor's input on an exemption. Please be assured that, once you have provided your new statement, the assessment of your exemption will not hold up the peer review process. 6. If the reviewer comments include a recommendation to cite specific previously published works, please review and evaluate these publications to determine whether they are relevant and should be cited. There is no requirement to cite these works unless the editor has indicated otherwise.

Reviewers' comments:

Reviewer's Responses to Questions

**Comments to the Author**

1. Is the manuscript technically sound, and do the data support the conclusions?

Reviewer #1: Partly

Reviewer #2: Yes

2. Has the statistical analysis been performed appropriately and rigorously? 

Reviewer #1: No

Reviewer #2: Yes

3. Have the authors made all data underlying the findings in their manuscript fully available?

Reviewer #1: No

Reviewer #2: Yes

4. Is the manuscript presented in an intelligible fashion and written in standard English?

Reviewer #1: Yes

Reviewer #2: Yes

5. Review Comments to the Author

**Reviewer #1:** Peer Review Comments

The manuscript "Hypergammaglobulinemia in treated and untreated people with HIV" reports a large cross-sectional analysis of serum IgG and inflammatory biomarkers across multiple HIV disease categories, including ART-suppressed participants. This decades spanning dataset offers valuable insights into residual immune activation despite viral suppression. While the authors have coined up a clinically relevant topic for the article, several revisions are needed to enhance clarity, methodological rigor, and consistency. Below are comments for the authors’ consideration.

MAJOR COMMENTS

1. The manuscript alternates between describing the study as retrospective and prospective. Please clarify the study's exact design and use consistent terminology in the Abstract, Methods, and Results sections. If this is a retrospective analysis of prospectively collected cohort data, state this clearly.

In lines 97–98, clarify whether some participants contributed multiple samples and how this was addressed in the analysis. If repeated measures were used, consider mixed effects or Generalized Estimating Equations (GEE) models, or select one sample per participant with sensitivity analyses. Additionally, in line 99, replace “One hundred sixty-six” with numerals.

2. The definition and ascertainment of primary HIV infection (PHI) require more detail. While PHI is defined as within 12 months of infection by the authors, the criteria used to determine this are not described. Specify whether the diagnosis was based on documented seroconversion, RNA+/Ab status, p24 antigen detection, or patient history, and address the possibility of late presenters being misclassified as PHI. Additionally, if authors could consider adding all category definitions in the article to a dedicated “Definitions” subsection.

3. Given that the data span from 1985 to 2023 and measurements were performed on different assay platforms, describe any calibration or harmonization procedures and assess whether platform or sampling era confounded the results. Sensitivity analyses restricted to uniform assay periods would strengthen the findings.

4. The authors state that the control group includes both PrEP and non‑PrEP individuals. It would be great if authors consider presenting subgroup analyses or adjusting for PrEP status in all control comparisons to avoid bias in defining what we may call “normal” reference ranges.

5. The statistical analysis involves numerous pairwise tests without adjustment for multiple comparisons. Apply false discovery rate or family‑wise error control and report effect sizes (e.g., median differences with 95% CIs) alongside p‑values. Clarify the handling of non‑normal data and log‑transformed variables.

6. Missing data for key biomarkers are noted in Table 1 and 2 captions. Authors could take time to state explicitly how missing values were handled, for instance, excluded, imputed, and assess whether the missingness was random.

7. In Table 2, summing the figures in column (n) gives a total of 892, which exceeds the stated sample size. Please clarify the actual study sample size and explain any discrepancies.

8. Figure 3 is mentioned in line 244 but is not included in the manuscript. Ensure all referenced figures are present, correctly numbered, and match their in‑text citations, and also the numbering must be sequentially ordered for both Tables and Figures in the article.

9. Some discussion statements, particularly those around lines 341-353, are not fully supported by the presented data. Revise to ensure interpretations are directly linked to results and acknowledge limitations in small subgroups.

10. Lines 41-43: The interpretation of inflammatory biomarker data could be expanded further. The statement “About one‑third of ART‑suppressed participants have elevated neopterin and β2‑microglobulin.” If authors could quantify the overlap with elevated IgG and explore predictors of persistent hypergammaglobulinemia using adjusted models, this could help with clarity on the data presentation in the article.

11. The Methods section and data analysis descriptions are not sufficiently detailed. Please expand these to clearly explain the analytical approach, including any statistical tests, adjustments, and handling of repeated measures.

12. The Swedish epidemic cohort reflects a male predominance, and the authors appropriately noted this. However, the finding that women had significantly higher IgG levels is interesting. Could this be explored further? Was this difference consistent across all CD4 strata and treatment groups? A brief comment on this would add depth.

13. Given the numerous factors influencing IgG levels (sex, treatment status, CD4 count, and inflammatory markers), a multivariate regression analysis would be highly valuable. This would help determine the independent predictors of IgG levels (and persistent hypergammaglobulinemia on ART) while controlling for other variables. The current univariate analyses are informative but could be significantly strengthened by a multivariate model.

14. Lines 314-322: The discussion on clinical implications is somewhat speculative. The suggestion that IgG could be a surrogate marker for B-cell dysfunction is interesting. Could the authors be more specific? For instance, in which clinical scenarios might measuring IgG be most useful (monitoring patients with incomplete immune reconstitution despite ART)? Also, please clarify the statement on lines 301-302; low albumin was only found in the OIM group, so it cannot be a general marker of HIV-related inflammation but rather of acute concomitant illness.

MINOR COMMENTS

1. Under the Abstract section, define all abbreviations at first use, for instance “human immunodeficiency virus (HIV)” and “immunoglobulin G (IgG)”. Apply this consistently for all abbreviations throughout the manuscript, including in tables and figures.

2. Line 133: "guidelines existing at the time of inclusion". Please specify the core principle, for instance, if authors considered a combination of at least two drugs from different classes. Additionally, what would be of value if guidelines could be cited.

3. Figure 2: The figure is helpful. Consider adding the p-values for the significant comparisons (HIV+ untreated and ART) directly to the figure or its legend for clarity.

4. Figure 3: The axis labels for Figure 3D should be "S-β2-microglobulin (mg/L)".

5. Line 275: "in 45 % and 37 %". This is fine, but for consistency, you could use the "%" symbol throughout (e.g., 45%).

6. Line 311: "mean age of 42 years". Could this be a typo in the results (Table 1) report? Please, authors, report the "median" instead.

7. Correct unit errors, for instance, (“ml/L” to “mg/L) for β2‑microglobulin, unless perhaps there were some conversions applied and typographical inconsistencies, for instance, “IgG -levels” to “IgG levels” (line 288) and standardize HIV‑1 vs HIV usage.

8. Ensure all tables and figures have self‑contained legends that define abbreviations, state reference limits, and specify comparison frameworks. Add 95% CIs to medians in tables where possible.

9. Where qualitative terms like “weak correlation” are used, include numeric rho and p‑values in text or figure captions.

10. When comparing results to other countries, provide possible explanations for differences (e.g., ART coverage, prophylaxis use, cohort characteristics) rather than stating comparability without context.

11. Check that all data in the text match the values in tables and figures and review the manuscript for typographical and formatting errors.

12. If applicable, include an accession or URL for the Swedish National Data Service in the Data Availability statement.

**Reviewer #2:** Title: Hypergammaglobulinemia in treated and untreated people with HIV

This paper by Vigmo et. al. is well-written and covers an important and under-investigated area of chronic HIV infection. The cohort is large and well-characterized, with a long follow-up period that includes both pre- and post-ART periods. The findings are clinically relevant and demonstrate persistent B-cell dysregulation (hypergammaglobulinemia) and immune activation even in virologically suppressed individuals. However, several points require clarification or expansion to strengthen the conclusions.

Major Comments

1. In lines 99-101, the authors mention that 166 participants provided samples twice and were thus included in both treated and untreated groups. This creates a significant issue of non-independence between the ART and the various untreated CD4 strata groups. The statistical comparisons between these groups are potentially confounded by this repeated sampling. The authors must clearly address this as a key limitation in the discussion. The statistical analysis should be clarified; were comparisons between groups treated as independent? If so, this is a methodological weakness. Consider using statistical methods that account for paired/repeated measures for these specific comparisons or focusing more on cross-group trends rather than direct pairwise tests between these non-independent groups.

2. In the methods section, Lines 98-101, Laboratory methods (lines 144-145), two different methods were used for IgG (nephelometry and immunoturbidimetry). Was there any harmonization or validation between these two methods across the long study period? This should be briefly mentioned to assure readers that measurements are comparable over time.

3. A major potential confounder for hypergammaglobulinemia is co-infection with Hepatitis C virus (HCV), which is common in PWH and a known cause of high IgG concentration. The authors did state this as a limitation (lines 327-329), but they do not provide any data. Ideally, the prevalence of HCV co-infection (and other chronic infections like Hepatitis B, Syphilis, and Epstein-Barr virus) should be reported for the key groups, especially the high IgG groups and the ART group. If data is available, a sensitivity analysis excluding HCV+ (or for any other chronic infectious disease) individuals would greatly strengthen the conclusion that the hypergammaglobulinemia is HIV-driven. If no data is available, this must be stated as a significant limitation.

Minor Comments

1. In line 167. The authors mention that ‘Partial correlation was performed when deemed appropriate.’ This is too vague. He authors need to specify what variables were adjusted for in the partial correlations (e.g., in Fig 3, were correlations adjusted for age, sex?).

2. In the results section, Table 1, the female percentage in the ‘With PrEP’ control group is 0% (0/45). This is a significant demographic difference from the ‘Without PrEP’ group (42.3% female) and the PWH groups. This should be noted in the text, since sex is later shown to have an influence on IgG levels.

6. PLOS authors have the option to publish the peer review history of their article (what does this mean?). If published, this will include your full peer review and any attached files.

Reviewer #1: No

Reviewer #2: No

---

## [Author Response · Author response to Decision Letter 1]

22 Nov 2025

Response to reviewers

“Hypergammaglobulinemia in treated and untreated people with HIV”

First, we wish to thank the two reviewers for their interest in our work and their constructive comments on the paper. We have revised the manuscript accordingly, and we address each comment below.

Reviewer 1

Major comments

Q1: The manuscript alternates between describing the study as retrospective and prospective. Please clarify the study's exact design and use consistent terminology in the Abstract, Methods, and Results sections. If this is a retrospective analysis of prospectively collected cohort data, state this clearly. In lines 97–98, clarify whether some participants contributed multiple samples and how this was addressed in the analysis. If repeated measures were used, consider mixed effects or Generalized Estimating Equations (GEE) models, or select one sample per participant with sensitivity analyses. Additionally, in line 99, replace “One hundred sixty-six” with numerals.

A1: Thank You for this valuable comment. We have clarified that the study is a cross-sectional analysis of prospectively collected data, and the term “retrospective” has been removed throughout the manuscript.

For the group comparisons in Table 2, no participant contributed more than one sample; each sample represents a unique subject, either in the ART group or one of the untreated groups, thus all samples are independent. This has been clarified in the Definitions section (lines 121-123) and in the captions of Table 1 and 2 (lines 189-192 and 219-222). Prompted by this remark, we also revised Figure 2, dividing the HIV-positive samples into ART-treated and untreated groups. Finally, “one hundred sixty-six” has been replaced with numerals.

Q2: The definition and ascertainment of primary HIV infection (PHI) require more detail. While PHI is defined as within 12 months of infection by the authors, the criteria used to determine this are not described. Specify whether the diagnosis was based on documented seroconversion, RNA+/Ab status, p24 antigen detection, or patient history, and address the possibility of late presenters being misclassified as PHI. Additionally, if authors could consider adding all category definitions in the article to a dedicated “Definitions” subsection.

A2: The diagnosis of primary HIV infection (PHI) was based on a combination of clinical symptoms and laboratory confirmation through seroconversion, nucleic acid testing or ELISA results. This information has been added to the Definitions section (lines 110-112). The combined use of clinical and laboratory data minimizes the risk of misclassification of infection stage. Further, definitions for all study categories have now been consolidated under a dedicated “Definitions” subsection, as suggested.

Q3: Given that the data span from 1985 to 2023 and measurements were performed on different assay platforms, describe any calibration or harmonization procedures and assess whether platform or sampling era confounded the results. Sensitivity analyses restricted to uniform assay periods would strengthen the findings.

A3: Thank You for raising this important point regarding the change in analytical methods for IgG measurement during the study period. When the analytical instrument was replaced, a comprehensive validation was performed, including verification of the reference interval. The laboratory has also participated continuously in two external quality assurance programs (Equalis and INSTAND). According to available validation and external quality control data, the transition from nephelometry to immunoturbidimetry did not result in any clinically relevant differences in IgG concentrations. This information has been added to the Laboratory methods section (lines 143-145).

Q4: The authors state that the control group includes both PrEP and non PrEP individuals. It would be great if authors consider presenting subgroup analyses or adjusting for PrEP status in all control comparisons to avoid bias in defining what we may call “normal” reference ranges.

A4: While HIV-negative controls using PrEP showed higher concentrations of β2-microglobulin and neopterin when compared to HIV-negative controls not using PrEP, there was no significant difference in IgG levels between these groups. This has now been stated in the Results section (lines 204-206). The observed biomarker differences between PrEP and non-PrEP controls have been described previously (Robertson et al, references 28 and 30) and this has now been addressed in the Results (lines 262-265) as well as in the Discussion (lines 351-355. We have also performed a subgroup analysis comparing HIV-negative controls with and without PrEP in all control comparisons and the overall results remained unchanged (lines 206-207).

Q5: The statistical analysis involves numerous pairwise tests without adjustment for multiple comparisons. Apply false discovery rate or family wise error control and report effect sizes (e.g., median differences with 95% CIs) alongside p values. Clarify the handling of non normal data and log transformed variables.

A5: In table 2, 32 pairwise tests were performed. Using Holm-Bonferroni, which is still conservative due to correlated tests, to correct for multiple inference does not change the status of any of the nominally significant tests. We agree that effect sizes are generally important, however, in this context we believe including them would make the table less readable. For this reason, we decided not to include these data. Regarding data distribution, non-normally distributed variables were log-transformed prior to analysis, as stated in the Statistical methods section (lines 165-175).

Q6: Missing data for key biomarkers are noted in Table 1 and 2 captions. Authors could take time to state explicitly how missing values were handled, for instance, excluded, imputed, and assess whether the missingness was random.

A6: Thank You for this comment. Regarding the missing data for inflammatory biomarkers (neopterin, β2-microglobulin, albumin), these were missing by random, e.g., when the remaining blood volumes were insufficient for the required assays. Hence, no systematic error was identified. Missing data were excluded pairwise in the statistical analyses. This information has been added to the captions of Table 1 and Table 2 (lines 193-194 and 224-225, respectively).

Q7: In Table 2, summing the figures in column (n) gives a total of 892, which exceeds the stated sample size. Please clarify the actual study sample size and explain any discrepancies.

A7: Thank You for noting this. The HIV-negative control group (n = 71) was subdivided into participants using PrEP (n = 45) and not using PrEP (n = 26). Thus, the total number of samples (n = 821) is correct. To clarify this, we have right-aligned the participant number for the HIV-negative subgroups (with and without PrEP) in Table 1 and Table 2. In addition, information on the total sample size has been added to the table legends (lines 189-192 and 219-222).

Q8: Figure 3 is mentioned in line 244 but is not included in the manuscript. Ensure all referenced figures are present, correctly numbered, and match their in text citations, and also the numbering must be sequentially ordered for both Tables and Figures in the article.

A8: Figure 3 was submitted along with the manuscript, but seems to be missing in the file presented to Reviewer 1.

Q9: Some discussion statements, particularly those around lines 341-353, are not fully supported by the presented data. Revise to ensure interpretations are directly linked to results and acknowledge limitations in small subgroups.

A9: Thank You for this comment. It appears there may have been a line displacement, as lines 341-353 contained the supporting figure captions rather than discussion text. We would appreciate clarification regarding which specific part(s) of the discussion Reviewer 1 is referring to, so that we can address the concern appropriately.

Q10: Lines 41-43: The interpretation of inflammatory biomarker data could be expanded further. The statement “About one third of ART suppressed participants have elevated neopterin and β2 microglobulin.” If authors could quantify the overlap with elevated IgG and explore predictors of persistent hypergammaglobulinemia using adjusted models, this could help with clarity on the data presentation in the article.

A10: We assume that the Reviewer is referring to lines 228-230 in the previous manuscript version. Instead of quantifying the overlap, we would like to refer to Figure 3, which appears to be missing from the manuscript provided to Reviewer 1. Here, correlation coefficients were explored for IgG regarding HIV-RNA, CD4+, neopterin, beta-2-microglobulin, albumin, and age in untreated and ART-suppressed HIV-positive participants.

Q11: The Methods section and data analysis descriptions are not sufficiently detailed. Please expand these to clearly explain the analytical approach, including any statistical tests, adjustments, and handling of repeated measures.

A11: Thank You for this valuable comment. The statistical section has been rewritten.

Q12: The Swedish epidemic cohort reflects a male predominance, and the authors appropriately noted this. However, the finding that women had significantly higher IgG levels is interesting. Could this be explored further? Was this difference consistent across all CD4 strata and treatment groups? A brief comment on this would add depth.

A12: We agree with the Reviewer that the gender differences in IgG levels warrant further discussion. We have therefore expanded the analysis of IgG levels in men and women within the untreated groups, and this is commented on in the Results section (lines 241-246). Gender differences in IgG levels in the general population as well as among HIV-negative controls have been addressed in the Discussion section (lines 329-336). Additionally, Figure 2 has been modified to present HIV-positive samples divided into ART-treated and untreated groups.

Q13: Given the numerous factors influencing IgG levels (sex, treatment status, CD4 count, and inflammatory markers), a multivariate regression analysis would be highly valuable. This would help determine the independent predictors of IgG levels (and persistent hypergammaglobulinemia on ART) while controlling for other variables. The current univariate analyses are informative but could be significantly strengthened by a multivariate model.

A13: We have performed a multivariable analysis, but did not include neopterin and β2-microglobulin, since we do not consider them as causal determinants to IgG, but rather correlated parallel outcomes to IgG that misleadingly attenuate the impact of the other predictors when included. The results from the multivariable analyses are reported in the Results section (lines 207-211).

Q14: Lines 314-322: The discussion on clinical implications is somewhat speculative. The suggestion that IgG could be a surrogate marker for B-cell dysfunction is interesting. Could the authors be more specific? For instance, in which clinical scenarios might measuring IgG be most useful (monitoring patients with incomplete immune reconstitution despite ART)? Also, please clarify the statement on lines 301-302; low albumin was only found in the OIM group, so it cannot be a general marker of HIV-related inflammation but rather of acute concomitant illness.

A14: We have revised the final sentence to make the interpretation of persistent hypergammaglobulinemia of ART-suppressed participants less speculative (lines 385-387).

We also agree that the observed hypoalbuminemia in the OIM study group is likely related to the concomitant infection or malignancy, rather than to HIV-infection itself. This has been clarified in the Discussion section (lines 355-357).

Minor comments

Q1: Under the Abstract section, define all abbreviations at first use, for instance “human immunodeficiency virus (HIV)” and “immunoglobulin G (IgG)”. Apply this consistently for all abbreviations throughout the manuscript, including in tables and figures

A1: All abbreviations have now been defined in the manuscripts as well as in the figure and table legends.

Q2: Line 133: "guidelines existing at the time of inclusion". Please specify the core principle, for instance, if authors considered a combination of at least two drugs from different classes. Additionally, what would be of value if guidelines could be cited.

A2: We have added citations for past and present Swedish treatment guidelines (lines 114-118).

Q3: Figure 2: The figure is helpful. Consider adding the p-values for the significant comparisons (HIV+ untreated and ART) directly to the figure or its legend for clarity.

A3: P values have been added. Further, we have chosen to divide the HIV-positive group into treated and untreated participants, since 166 participants provided two samples and hence were included in both groups.

Q4: Figure 3: The axis labels for Figure 3D should be "S-β2-microglobulin (mg/L)".

A4: Hyphens have been added consistently in tables and figures.

Q5: Line 275: "in 45 % and 37 %". This is fine, but for consistency, you could use the "%" symbol throughout (e.g., 45%).

A5: The gaps between numbers and percentage signs have been removed.

Q6: Line 311: "mean age of 42 years". Could this be a typo in the results (Table 1) report? Please, authors, report the "median" instead.

A6: “Mean” has been changed to “median”. Moreover, we have reported the median age for untreated and treated participants, respectively.

Q7: Correct unit errors, for instance, (“ml/L” to “mg/L) for β2 microglobulin, unless perhaps there were some conversions applied and typographical inconsistencies, for instance, “IgG -levels” to “IgG levels” (line 288) and standardize HIV 1 vs HIV usage.

A7: These inconsistencies have been corrected.

Q8: Ensure all tables and figures have self contained legends that define abbreviations, state reference limits, and specify comparison frameworks. Add 95% CIs to medians in tables where possible.

A8: The legends for all tables and figures have been reviewed to ensure they are self-contained, with definitions of abbreviations and clear descriptions of comparisons frameworks. Reference intervals have been added to the legends of Table 2, Fig 2, Fig 3 and S2 Fig. We chose not to add 95 % CI, as we believe presenting median (IQR) values provide a clearer and more accessible summary of the data.

Q9: Where qualitative terms like “weak correlation” are used, include numeric rho and p values in text or figure captions.

A9: Numeric rho and P values are shown in Figure 3, which unfortunately seems to be missing for Reviewer 1.

Q10: When comparing results to other countries, provide possible explanations for differences (e.g., ART coverage, prophylaxis use, cohort characteristics) rather than stating comparability without context.

A10: No comparisons between countries have been made in the manuscript.

Q11: Check that all data in the text match the values in tables and figures and review the manuscript for typographical and formatting errors.

A11: The manuscript has been reviewed for typographical and formatting errors.

Q12: If applicable, include an accession or URL for the Swedish National Data Service in the Data Availability statement.

A12: The anonymized dataset for the present study has been uploaded to the Swedish National Data Service with restricted access due to participant confidentiality. The DOI is stated in the Data availability statement, line 391.

Reviewer 2

Major comments

Q1: In lines 99-101, the authors mention that 166 participants provided samples twice and were thus included in both treated and untreated groups. This creates a significant issue of non-independence between the ART and the various untreated CD4 strata groups. The statistical comparisons between these groups are potentially confounded by this repeated sampling. The authors must clearly address this as a key limitation in the discussion. The statistical analysis should be clarified; were comparisons between groups treated as independent? If so, this is a methodological weakness. Consider using statistical methods th

---

## [Decision Letter · Decision Letter 1]

17 Feb 2026

PONE-D-25-36523R1Hypergammaglobulinemia in treated and untreated people with HIVPLOS One

Dear Dr. Vigmo,

Thank you for submitting your manuscript to PLOS ONE. After careful consideration, we feel that it has merit but does not fully meet PLOS ONE’s publication criteria as it currently stands. Therefore, we invite you to submit a revised version of the manuscript that addresses the points raised during the review process.

**Please kindly respond to reviewer 3 and resubmit a revised version.**

We look forward to receiving your revised manuscript.

Kind regards,

Sepiso K. Masenga, PhD

Academic Editor

PLOS One

Journal Requirements:

Reviewers' comments:

Reviewer's Responses to Questions

**Comments to the Author**

1. If the authors have adequately addressed your comments raised in a previous round of review and you feel that this manuscript is now acceptable for publication, you may indicate that here to bypass the “Comments to the Author” section, enter your conflict of interest statement in the “Confidential to Editor” section, and submit your "Accept" recommendation.

Reviewer #1: All comments have been addressed

Reviewer #3: All comments have been addressed

2. Is the manuscript technically sound, and do the data support the conclusions?

Reviewer #1: Yes

Reviewer #3: Yes

3. Has the statistical analysis been performed appropriately and rigorously? 

Reviewer #1: Yes

Reviewer #3: Yes

4. Have the authors made all data underlying the findings in their manuscript fully available?

Reviewer #1: Yes

Reviewer #3: Yes

5. Is the manuscript presented in an intelligible fashion and written in standard English?

Reviewer #1: Yes

Reviewer #3: Yes

6. Review Comments to the Author

Reviewer #1: The authors have done a great job addressing all the concerns which were raised. Great work on this manuscript.

Reviewer #3: Peer Review Report

Manuscript Title: Hypergammaglobulinemia in treated and untreated people with HIV

I thank the editorial team for the opportunity to review this revised manuscript. I commend the authors for their thoughtful engagement with the previous review comments; the amendments have certainly strengthened the paper. However, to ensure the highest level of clarity and rigor, I have a few remaining points that require the authors' attention before the manuscript can be considered for acceptance.

Specific Comments:

1. Clarity of Statistical Reporting (Major Point): In Figure 3, the presentation of the correlation coefficients requires revision for consistency and clarity. Specifically, the notation for the correlation coefficients (rp) should be standardized to use a decimal point, not a comma. For instance:

- In Fig 3C, please clarify whether the values are `rp = 0.076, p = 0.102` and `rp = 0.074, p = 0.263`.

- Similarly, in Fig 3E, the value should be presented as `rp = -0.277, p < 0.001`.

Ensuring this standardized formatting is crucial for unambiguous data interpretation.

2. Discussion of Demographic Limitations (Major Point): The authors have appropriately addressed several reviewer comments and have noted the male predominance within the Swedish epidemic cohort. However, a critical demographic imbalance persists between the control groups that warrants explicit acknowledgment as a study limitation.

- The ‘With PrEP’ control group is exclusively male (0% female, 0/45), which stands in stark contrast to the 'Without PrEP' control group (42.3% female) and the cohorts of people with HIV (PWH).

- This significant difference introduces a potential source of demographic bias that could influence the comparative analyses of immunoglobulin levels. I strongly recommend that the authors explicitly add a statement to the Discussion or Limitations section acknowledging this imbalance. A cautionary note on how this might affect the generalizability or interpretation of the findings related to the PrEP-naïve control group would significantly strengthen the manuscript's scientific rigor and transparency.

7. PLOS authors have the option to publish the peer review history of their article (what does this mean?). If published, this will include your full peer review and any attached files.

Reviewer #1: No

Reviewer #3: **Yes:** David Chisompola

---

## [Author Response · Author response to Decision Letter 2]

18 Mar 2026

Response to Reviewer 3:

Q1: Clarity of Statistical Reporting (Major Point): In Figure 3, the presentation of the correlation coefficients requires revision for consistency and clarity. Specifically, the notation for the correlation coefficients (rp) should be standardized to use a decimal point, not a comma. For instance: In Fig 3C, please clarify whether the values are `rp = 0.076, p = 0.102` and `rp = 0.074, p = 0.263`. Similarly, in Fig 3E, the value should be presented as `rp = -0.277, p < 0.001`. Ensuring this standardized formatting is crucial for unambiguous data interpretation.

A1: Thank you for pointing this out. The comma has been corrected to a decimal point in the revised Figure 3.

Q2: Discussion of Demographic Limitations (Major Point): The authors have appropriately addressed several reviewer comments and have noted the male predominance within the Swedish epidemic cohort. However, a critical demographic imbalance persists between the control groups that warrants explicit acknowledgment as a study limitation.

- The ‘With PrEP’ control group is exclusively male (0% female, 0/45), which stands in stark contrast to the 'Without PrEP' control group (42.3% female) and the cohorts of people with HIV (PWH).

- This significant difference introduces a potential source of demographic bias that could influence the comparative analyses of immunoglobulin levels. I strongly recommend that the authors explicitly add a statement to the Discussion or Limitations section acknowledging this imbalance. A cautionary note on how this might affect the generalizability or interpretation of the findings related to the PrEP-naïve control group would significantly strengthen the manuscript's scientific rigor and transparency.

A2: We agree that the gender imbalance in the HIV-negative control group on PrEP represents a potential limitation of the study. To further address this, we have added statements in the Discussion section (lines 332-334 and 380-382), acknowledging this imbalance and its potential influence on the generalizability of the findings.

---

## [Decision Letter · Decision Letter 2]

30 Apr 2026

Hypergammaglobulinemia in treated and untreated people with HIV

PONE-D-25-36523R2

Dear Dr. Vigmo,

We’re pleased to inform you that your manuscript has been judged scientifically suitable for publication and will be formally accepted for publication once it meets all outstanding technical requirements.

Kind regards,

Mattia Trunfio, M.D., Ph.D

Academic Editor

PLOS One

Additional Editor Comments (optional):

Interesting work!

Reviewers' comments:

Reviewer's Responses to Questions

**Comments to the Author**

1. If the authors have adequately addressed your comments raised in a previous round of review and you feel that this manuscript is now acceptable for publication, you may indicate that here to bypass the “Comments to the Author” section, enter your conflict of interest statement in the “Confidential to Editor” section, and submit your "Accept" recommendation.

Reviewer #1: All comments have been addressed

Reviewer #3: All comments have been addressed

2. Is the manuscript technically sound, and do the data support the conclusions?

Reviewer #1: Yes

Reviewer #3: Yes

3. Has the statistical analysis been performed appropriately and rigorously? 

Reviewer #1: Yes

Reviewer #3: Yes

4. Have the authors made all data underlying the findings in their manuscript fully available?

Reviewer #1: Yes

Reviewer #3: Yes

5. Is the manuscript presented in an intelligible fashion and written in standard English?

Reviewer #1: Yes

Reviewer #3: Yes

6. Review Comments to the Author

Reviewer #1: Great work that has been presented by the authors. The reviewers concerns were addressed. One more vital concern to address; the attached uploaded supplementary file "S1 Table", is not clear what it represents: 1. It has no tag/title 2. The first column, equally the title is not defined.

If authors could kindly attend to this matter. Best regards.

Reviewer #3: I thank the authors for the thoughtful and comprehensive responses to the reviewers’ comments. The revisions have substantially strengthened the manuscript in terms of clarity, rigor, and overall presentation. The authors have adequately addressed the concerns raised, and the manuscript has improved considerably as a result.

I recommend acceptance of the manuscript for publication.

7. PLOS authors have the option to publish the peer review history of their article (what does this mean?). If published, this will include your full peer review and any attached files.

Reviewer #1: No

Reviewer #3: No

---

## [Editor Report · Acceptance letter]

PONE-D-25-36523R2

PLOS One

Dear Dr. Vigmo,

I'm pleased to inform you that your manuscript has been deemed suitable for publication in PLOS One. Congratulations! Your manuscript is now being handed over to our production team.

Kind regards,

on behalf of

Dr. Mattia Trunfio

Academic Editor

PLOS One